# Systematic Review and Meta-Analysis of Metabolic Syndrome and Its Components in Latino Immigrants to the USA

**DOI:** 10.3390/ijerph20021307

**Published:** 2023-01-11

**Authors:** Talita Monsores Paixão, Liliane Reis Teixeira, Carlos Augusto Ferreira de Andrade, Debora Sepulvida, Martha Martinez-Silveira, Camila Nunes, Carlos Eduardo Gomes Siqueira

**Affiliations:** 1Center of Studies of Worker Health and Human Ecology, National School of Public Health Sergio Arouca, Oswaldo Cruz Foundation, Rio de Janeiro 21041-210, Brazil; 2Department of Epidemiology and Quantitative Methods in Health, National School of Public Health Sergio Arouca, Oswaldo Cruz Foundation, Rio de Janeiro 21041-210, Brazil; 3Brazilian Worker Center, Boston, MA 02134, USA; 4Gonçalo Moniz Institute, Oswaldo Cruz Foundation, Bahia 40296-710, Brazil; 5Fluminense Federal Institute of Education Science and Technology, Campos dos Goytacazes 28030-130, Brazil; 6School for the Environment, University of Massachusetts Boston, Boston, MA 02125, USA

**Keywords:** metabolic syndrome, Latinos, immigrants, USA

## Abstract

The Metabolic Syndrome (MetS) is an increasingly prevalent condition globally. Latino populations in the USA have shown an alarming increase in factors associated with MetS in recent years. The objective of the present systematic review was to determine the prevalence of MetS and its risk factors in immigrant Latinos in the USA and perform a meta-analysis of those prevalence. The review included cross-sectional, cohort, or case–control studies involving adult immigrant Latinos in the USA, published during the period 1980–2020 in any language. Studies involving individuals who were pregnant, aged <18 years, immigrant non-Latinos, published outside the 1980–2020 period, or with other design types were excluded. The Pubmed, Web of Science, Embase, Lilacs, Scielo, and Google Scholar databases were searched. The risk of bias was assessed using the checklists of the Joanna Briggs Institute. The review included 60 studies, and the meta-analysis encompassed 52 studies. The pooled prevalence found for hypertension, diabetes, general obesity, and abdominal obesity were 28% (95% Confidence Interval (CI): 23–33%), 17% (95% CI: 14–20%), 37% (95% CI: 33–40%), and 54% (95% CI: 48–59%), respectively. The quality of the evidence of the primary studies was classified as low or very low. Few studies including immigrants from South America were identified. Further studies of those immigrants are needed due to the cultural, dietary, and language disparities among Latin American countries. The research protocol was registered with the Open Science Framework (OSF).

## 1. Introduction

The last few decades have seen a shift in the morbidity–mortality profile of the population globally, first in developed countries and then in developing nations. The epidemiological transition theory was first put forward by Omran [1] in 1971. According to this theory, the transition stems from long-term changes in patterns of mortality, and illnesses caused by infectious diseases are gradually giving way to an increased occurrence of chronic–degenerative diseases as the leading cause of morbidity and death.

According to Omran [1], the transition from the predominance of infectious–contagious diseases to chronic diseases involves three main groups of determinants: ecobiological determinants, socioeconomic, political and cultural determinants, and public health determinants.

There is currently a steady increase in cardiovascular disease, diabetes, cancers, and obesity. Regarding obesity, it is now considered a global epidemic, with the United States (USA) numbering as one of the first countries to show that rising obesity was becoming epidemic [2,3]. Obesity, together with hypertension, diabetes, and dyslipidemia, is one of the main risk factors for Metabolic Syndrome (MetS), a condition that has been widely described and whose prevalence tends to accompany rises in obesity [4,5].

The MetS was first described by Reaven [6]. The National Cholesterol Education Program (NCEP) [7] later produced the first standardized definition of MetS, updated in 2005 by Grundy et al. [8].

Currently, the diagnostic criterion for MetS is the co-occurrence of three or more of the following factors: high waist circumference (≥102 cm in men and ≥88 cm in women) or body mass index (BMI) > 30 kg/m^2^ (WHO [9]); high triglycerides (≥150 mg/dL or 1.7 mmol/L) or treatment for high triglycerides; low hide density lipoprotein (HDL) cholesterol (<40 mg/dL or 1.03 mmol/L in men and <50 mg/dL or 1.3 mmol/L in women) or treatment for low HDL cholesterol; high blood pressure (systolic ≥130 mm/Hg and diastolic ≥85 mm/Hg) or treatment for systemic arterial hypertension; and high fasting blood glucose (≥100 mg/dL) or treatment for type 2 diabetes.

In a 2020 study of USA adults [10], the authors found an MetS prevalence of 61.6% in obese individuals, 33.2% in overweight subjects, and 8.6% in adults with normal weight, highlighting the strong link between obesity and MetS. Despite the increasing prevalence of obesity in the general population of the USA over recent years, doubling between 1980 and 2010, this rate has now stabilized at approximately 35%. However, this rise remains alarming among minority populations, especially Latinos and black non-Latinos [5,11]. Similar increases for hypertension, diabetes, and high cholesterol have been documented in Latinos between 2001 and 2020 [12].

The Latino population In the USA is the largest and fastest growing racial/ethnic group in the country. According to USA Census data [13], the Latino population reached a total of 62.5 million people in 2021, a 19% increase from the 50.5 million Latinos in 2010. In California and Texas, Latinos became the largest racial/ethnic group. Florida, Arizona, and New York also had significant increases in the Latino population between the 2010 and 2021 Censuses.

Furthermore, the proportion of Latinos who speak English proficiently and who have a college education have grown since the 2010 Census [13]. However, Latinos in the USA are a very heterogeneous group [14] due to their cultural and socioeconomic differences, which demonstrate important variation according to their country of origin or ancestry. The average family income by Latino nationality, for example, shows that USA Latinos of Argentinian origin have the highest average income (USD 68,000) compared with the average for the USA Latino population (USD 49,010); those of Mexican origin have an average income of USD 49,000, while those of Honduran origin have the lowest average income (USD 41,000) [13].

Since the USA is one of the countries with the highest level of social disparities [15], the socioeconomic and educational disparities found among Latino populations according to nationality or ancestry must be considered in any public health studies of USA Latinos. Similar concerns should also apply to cultural differences between USA Latinos, because the fact that most USA Latinos are fluent in Spanish is not the only cultural aspect to consider regarding the determinants of MetS.

High rates of the factors associated with MetS in immigrant Latinos in the USA have been observed in previous studies, particularly among Mexicans [14,15,16]. Two systematic reviews [17,18] found increased obesity in Latinos, attributing this rise to higher levels of acculturation and residence in the USA, as well as immigrant generational and nativity status. The increase in obesity and in the other diseases defining MetS seen among immigrant populations is often associated with the nutritional transition occurring in developing countries, together with shifts in dietary habits and levels of physical activity, highlighted in both reviews [16,17]. For Brazil, a country which has been undergoing a process of nutritional transition since the 1970s [18], data from the Vigitel Brasil system [19] reveal an increase in the risk factors for chronic diseases, particularly overweight and obesity, indicating an increased risk of developing MetS.

Immigrants from developing countries undergoing a process of nutritional transition who emigrate to developed countries with a highly obesogenic environment are subject to an acceleration in the process of nutritional transition, favoring the development of obesity and its associated factors [17]. Nonetheless, when assessing such factors, it is important that cultural, language, economic, and social characteristics of the immigrants be taken into account, as well as origin.

Another key point is the very high likelihood that Brazilian immigrants were not included in samples of most studies conducted in the USA, owing to the classification adopted by the USA Census for the Latino population. Individuals with Cuban, Mexican, or Puerto Rican background/origin are usually classified as Hispanic or Latinos, whereas those from other Central and South American countries are identified as “others with Hispanic, Latino, or Spanish origin” [20]. Thus, the generalization for Brazilian immigrants of health information identified in studies of immigrants in the USA, using the term “Latino” as a reference, is not appropriate unless one can ascertain that Brazilians were included in the data sources.

Given the growth in diseases associated with MetS in immigrant Latinos in the USA and the epidemiological transition that has taken place and is still underway globally, in which chronic diseases feature as the leading cause of morbidity–mortality, identifying the publications involving this population to accurately determine the prevalence of MetS and its components is crucial.

However, the vast majority of publications on health conditions of immigrant Latinos in the USA address only immigrants from Mexico and Central America. Thus, how or whether South American immigrants are included in those studies remains unclear. Taking into account the cultural and economic heterogeneity that exists across Latin America, where Brazil has different language, culture, customs, and eating habits from other Spanish-speaking nations, analyses centering on the Brazilian population are important. 

Therefore, the objective of our systematic review was to determine the prevalence of MetS and its risk factors among immigrant Latinos in the USA and perform a meta-analysis of those rates. This review also sought to ascertain which Latino groups are included and the extent to which Brazilians feature in those published studies.

## 2. Materials and Methods

### 2.1. Protocol

Our systematic review was carried out according to the PICO (Population, Intervention/Exposure, Comparator(s)/Control and Outcomes) strategy, where:Population: Immigrants from Latin America, residing in the USA, aged >18 years;Intervention/Exposure: Immigration;Comparator(s)/control: US-born Latino population;Outcomes: MetS and/or its components (primary) and sleep disorders (secondary).

The study protocol was based on the criteria established by PROSPERO, an International Prospective Register of Systematic Reviews, and registered with the Open Science Framework (OSF) before starting the review (https://doi.org/10.17605/OSF.IO/JFM7G (accessed on 7 January 2023)). The systematic review yielded sufficient data to perform a meta-analysis of prevalence, which was subsequently incorporated into the review. The systematic review was reported according to Preferred Reporting Items for Systematic Reviews and Meta-Analyses (PRISMA) 2020 statement [21] (Appendix A).

### 2.2. Definition of Disease and Disease Codes

The primary outcomes included in our review were the MetS and/or its risk factors: high blood pressure or diagnosis/treatment of arterial hypertension; high fasting blood glucose or diagnosis/treatment of type 2 diabetes mellitus; low HDL-c or treatment for low HDL-c; high triglycerides or treatment for high triglycerides; and high waist circumference or BMI > 30 kg/m^2^ [7,9]. Sleep disorders were the secondary outcomes.

Based on the codes of the International Statistical Classification of Diseases and Related Health Problems (ICD-10) [22], the relevant categories for the review were: E88.8—other specified metabolic disorders (MetS), I10—essential hypertension (primary), E78—disorders of lipoprotein metabolism and other lipidemias, E11—type 2 diabetes mellitus, E66—obesity, and G47—sleep disorders.

### 2.3. Search Strategy

The search strategy was devised to include all important descriptors needed to retrieve the relevant studies for the review (Appendix A).

The PubMed, Web of Science, Embase, Lilacs, and Scielo databases were searched, along with Google Scholar as a complementary source, with no restrictions regarding language of publication. The search strategy was applied for the last time on 3 June 2020. 

### 2.4. Study Selection

Studies that met the inclusion criteria listed below were selected for inclusion in the review: Cross-sectional, cohort, or case–control type study;Involving adults (>18 years);Investigating immigrant Latinos residing in the USA;Published during the period 1980–2020 in any language.

Studies involving pregnant women, immigrants from countries other than Latin America, children and adolescents (<18 years), published outside the 1980–2020 period or based on data collected prior to 1980, qualitative or case studies, and those with non-original data were excluded from the review.

Two reviewers (TP, DS) independently screened titles and abstracts of the articles retrieved using the search strategy by applying the Rayyan [23] app and Excel^®^ spreadsheets. The full texts of the articles selected were then obtained, reviewed, and categorized as “included” or “excluded” by the same reviewers (TP, DS) in a double-blind analysis, according to the predefined inclusion criteria. The reasons for exclusion were documented using the Rayyan app [23]. At both stages, any discrepancies were checked by a third reviewer (LT).

### 2.5. Data Extraction

For all articles included in the previous stage, three reviewers (TP, DS, CN) extracted the data independently using an extraction spreadsheet previously created and validated by the research team, containing the following parameters: Study characteristics: authors, year of publication, author affiliations, e-mail of corresponding author, and study title;Study population: total number of participants, total number of women and men, whether study stratified analyses by Brazilian immigrants, immigrant group studies, age, country of origin, comparative group, and length of residence in the USA;Study design: design type and study period;Exposures;Other risk factors: work, shift work, documented or otherwise, and health insurance;Outcomes: MetS, hypertension or high blood pressure, type 2 diabetes or high fasting glucose, low HDL-c, high triglycerides, abdominal obesity or BMI > 30 kg/m^2^ (primary), and sleep disorders (secondary).

### 2.6. Risk of Bias

The risk of bias of the studies included was assessed using the checklists of the Joanna Briggs Institute (JBI) titled “Critical Appraisal Checklist for Analytical Cross-Sectional Studies, Checklist for Case-Control Studies, and Checklist for Cohort Studies” [24]. Located within the Faculty of Health Sciences of Adelaide University, the JBI is an international research and development organization world-renowned for evidence-based healthcare [24].

Given that systematic reviews represent a summary of core evidence, the JBI developed processes for the critical evaluation and synthesis of evidence to aid decision-making in health. The eight domains of risk of bias included in the JBI Critical Appraisal Checklist for Analytical Cross-Sectional Studies are: (i) clearly defined criteria for inclusion; (ii) description of study subjects and setting; (iii) valid and reliable measurement of exposure; (iv) use of objective, standard criteria; (v) identification of confounding factors; (vi) strategies for dealing with confounding factors; (vii) measurement of outcomes; and (viii) appropriate statistical analysis [24].

The JBI Critical Appraisal Checklist for Case Control Studies has 10 domains: (i) comparability of the groups; (ii) appropriateness of matching of cases and controls; (iii) criteria used for identification of cases and controls; (iv) valid and reliable measurement of exposure; (v) identification of confounding factors; (vi) identification of confounding factors; (vii) strategies for dealing with confounding factors; (viii) measurement of outcomes; (ix) length of exposure period of interest; and (x) appropriate statistical analysis [24].

The JBI Critical Appraisal Checklist for Cohort Studies has 11 domains: (i) similarity between groups and recruitment in the same population; (ii) similarity in measurement of exposures in assigning participants to the exposed and unexposed groups; (iii) valid and reliable measurement of exposure; (iv) identification of confounding factors; (v) strategies for dealing with confounding factors; (vi) outcome-free groups or participants at study baseline or exposure; (vii) measurement of outcomes; (viii) appropriate follow-up time; (ix) description of conclusion of follow-up or reason for loss to follow-up; (x) use of strategies to address incomplete follow-up; and (xi) appropriate statistical analysis [24]. 

For each study included the questions from the respective JBI Checklist were answered with a “yes”, “no”, “unclear”, or “not applicable”. The general risk of bias for each study was determined according to the following cut-off points [24]: ≥70% “yes” answers: low risk of bias;50–69% “yes” answers: moderate risk of bias;<50% “yes” answers: high risk of bias.

Three of the study’s reviewers (TP, DS, CN) independently rated the risk of bias for each study included in the review, and a fourth reviewer (LT) settled any disagreements or discrepancies.

### 2.7. Analysis and Presentation of Results

The results obtained in the systematic review were first presented as a map of evidence, in which all articles reviewed were included according to study design, number of participants, characteristics of population, study site, and outcome for later analysis and discussion of findings. 

A meta-analysis of the prevalence of risk factors for the MetS was then conducted. The clinical heterogeneity of studies that had the same outcome was assessed independently by two authors, with a third author settling any disagreements. Since studies assessed many different, yet related, effects, they were pooled using the inverse variance technique employing random-effect models. The software Stata version 15.0 was used for statistical analysis. The I^2^ statistic was used to analyze the statistical heterogeneity of the studies. Pooled effect estimates were presented on the basis of the meta-analysis, even when the statistical heterogeneity was high (I^2^ > 70%), because high levels of statistical heterogeneity were expected. A subgroup and sensitivity analysis was performed comparing immigrant Latinos with US-born Latinos—including sample size, extreme estimates, and study design for the outcomes arterial hypertension or high blood pressure, type-2 diabetes mellitus or high blood glucose, and general obesity—to check their influence on the pooled estimate. We could not conduct sensitivity analysis for the other outcomes due to the limited number of articles included in the meta-analysis.

The quality of the evidence was analyzed using the assessment criteria of the Grading of Recommendations Assessment, Development and Evaluation (GRADE) system, the most widely used instrument for assessing the quality of evidence of studies included in systematic reviews [25]. GRADE differs from other evaluation instruments by separating quality of evidence and strength of recommendation, rating quality of evidence for each outcome of interest, and allowing the level of quality of evidence of observational studies to be rated up if three criteria are met. The latter are large effect size, presence of dose–response gradient, and residual confounding factors, which increase confidence in estimates [25]. Forest plots were used in the meta-analyses for outcomes rated as very low quality of evidence, although combined estimates are not shown. The results of the synthesis of the evidence are shown in the summary table of findings.

## 3. Results

### 3.1. Study Selection

Of the 2497 unique studies retrieved after applying the search strategy, 60 studies met the criteria for inclusion in the systematic review. Of this total, 52 studies were subsequently included in the meta-analysis, according to the model established by the PRISMA statement [21] (Figure 1). The eight studies excluded from the meta-analysis lacked the parameters required to estimate prevalence.

### 3.2. Study Characteristics

The total number of participants from all studies included in the review was 2,709,490. Most studies had a cross-sectional design (55) [15,16,26,27,28,29,30,31,32,33,34,35,36,37,38,39,40,41,42,43,44,45,46,47,48,49,50,51,52,53,54,55,56,57,58,59,60,61,62,63,64,65,66,67,68,69,70,71,72,73,74,75,76,77,78], followed by four cohort studies [14,79,80,81] and one case–control [82] study. Only two studies stratified Brazilian immigrants in their analyses [30,47]. Regarding country of origin of the immigrants studied, 14 studies involved Mexicans only [15,26,27,42,43,52,64,65,74,75,76,77,79,81], 11 Mexicans and other Latinos [16,33,34,36,49,51,54,61,63,66,67], 10 investigated immigrants from South and Central America (Brazil [30,47], Haiti [30,32], El Salvador [30,40], Colombia [30], Guatemala [30,40], Dominican Republic [30,46], Honduras [30,40], Peru [40], Bolivia [40], Puerto Rico [32,46], Mexico [46], Guyana [82] and other unspecified countries [32,35,38,39,45,46,82]), and 7 studied immigrants from Central America (Dominican Republic [69,72,80], Puerto Rico [69,72,80], Cuba [69,72], Mexico [72,78], El Salvador [78], Guatemala [78], Honduras [78], Haiti [56,59], and other unspecified countries [37,69,80]).

Participant age ranged from 0 to 91 years. Only one study included children and adolescents in its sample [56], age group 0 to ≥75 years. However, the analysis was stratified by age group, allowing data for subjects aged >18 years to be extracted.

Length of residence of the immigrants in the USA was assessed in 36 studies. Of these publications, 12 stratified length of residence into <10 years and >10 years [26,28,32,34,35,38,39,42,58,70,78,81]. The other studies included immigrants with <10 years of residence in the USA [30,65], <15 and >15 years [29,44,53,54,56], <20 and >20 years [41,50,66,74,80], or <25 and >25 years of residence [14]. Eleven studies [16,40,47,51,52,55,61,64,67,75,77] reported only average years of residence.

The exposures assessed in the studies were grouped under four categories related to immigration; immigrant; health, diet and lifestyle; and community. The distribution of exposures for the different categories is presented in Table 1.

In total, 17 articles analyzed work-related factors [16,31,33,39,41,43,44,46,47,53,55,61,64,66,67,68,70] and 9 described participant occupations, of which 4 involved rural or agricultural workers [31,33,61,70], 2 taxi/for-hire vehicle drivers, 1 professional/management and support services workers [44], and 1 female homemakers and others [66], while 1 encompassed a broad group of workers [16], dividing participants into: skilled professional, semiskilled white-collar, clerical, semiskilled blue-collar, unskilled service, laborer, or farmworker; homemaker; and unemployed or student. Four studies addressed the issue of immigrant documentation in their results [43,46,61,76] and 19 included information on holding health insurance or otherwise [28,31,33,36,37,39,40,45,46,49,54,55,56,57,63,68,75,78,80]. The characteristics of the studies included in the review are presented in Table 2.

### 3.3. Risk of Bias

For the 55 cross-sectional studies included in the systematic review, the overall scoring of risk of bias ranged from 25.5 to 100% for the eight domains of the JBI Critical Appraisal Checklist for Analytical Cross-Sectional Studies (Table 3). Most of the studies (50) had low risk of bias, three had moderate risk, and two had high risk of bias (Table 3).

Of the four cohort studies included and rated using the JBI Critical Appraisal Checklist for Cohort Studies (Table 4), two had low risk and two had moderate–high risk. The single case–control study included had low risk of bias according to the rating determined on the JBI Critical Appraisal Checklist for Case–Control Studies (Table 5).

### 3.4. Findings

#### 3.4.1. Outcomes for Latinos in the USA

Of the primary outcomes observed in the present review, 36 studies investigated hypertension [14,15,16,28,31,33,38,39,40,44,46,48,49,50,53,54,55,56,57,58,59,60,61,62,63,64,65,67,68,69,70,74,77,78,81,82], 34 assessed type 2 diabetes mellitus [14,15,16,31,33,34,36,37,38,39,40,43,44,46,50,52,54,56,57,59,60,61,63,64,65,67,68,69,70,75,77,78,80,82], 40 obesity (BMI ≥ 30 kg/m^2^) [14,15,16,26,27,29,30,31,32,33,34,35,37,40,41,42,43,44,46,47,50,51,52,54,58,60,61,63,64,65,66,67,70,71,72,73,74,76,78,79], 8 abdominal obesity (waist circumference) [15,26,39,60,64,65,72,77], 7 low cholesterol HDL [15,37,40,45,60,65,77], 7 high triglycerides [15,34,40,45,60,65,77], and 7 MetS [16,40,60,64,65,77]. Of the total studies reviewed, 21 analyzed at least three factors of MetS [14,15,16,33,34,37,39,40,44,46,50,54,60,61,63,64,65,67,70,77,78]. With regard to secondary outcomes, only two studies assessed sleep duration [34,44] (Table 6).

##### Arterial Hypertension or High Blood Pressure for Latinos in the USA

Of the 36 studies that assessed arterial hypertension, 24 measured blood pressure [14,15,28,31,33,38,39,40,48,50,53,57,59,60,61,63,64,65,67,69,74,77,78,82], 11 collected information based on self-reports [16,44,46,49,54,56,58,62,68,70,81], and 1 employed both methods [55].

The prevalence of arterial hypertension or high blood pressure in the studies reviewed in Latinos was 28% (95% CI: 23%-33%, I^2^: 99.6%). Of the studies included in the meta-analysis, 23 were heterogeneous (Figure 2). Within this group, some studies [16,39,50,53,68,70,78,82] differed more with respect to population size, measurement approach (objective or subjective), design (one case–control study) and characteristics of the population (ethnicity, age, and country of origin).

Additionally, a meta-analysis of the prevalence of arterial hypertension or high blood pressure for non-US-born Latinos (immigrants) and US-born Latinos was carried out (Figure 3). We found slightly higher prevalence for US-born Latinos (32% (95% CI: 19–45%, I^2^: 99.7%)), than for immigrant Latinos (28% (95% CI: 22–33%, I^2^: 99.3%)). The seven studies of immigrant Latinos and four of US-born Latinos included in the meta-analysis were heterogeneous. Greater differences were identified in two studies of immigrant Latinos [39,54] and two of US-born immigrants [14,54], possibly explained by population size, study design, or outcome measurement method.

##### Type 2 Diabetes Mellitus or High Blood Glucose for Latinos in the USA

Of the 34 studies which evaluated type 2 diabetes mellitus or high blood glucose, 21 analyzed blood samples to determine glucose levels [14,15,31,34,37,38,39,40,43,50,52,57,59,60,61,63,64,65,77,80,82], while 13 collected self-report information in interviews [16,33,36,44,46,54,56,67,68,69,70,75,78].

A total of 24 studies were included in the meta-analysis of the prevalence of type 2 diabetes mellitus or high blood glucose (Figure 4), although those were heterogeneous. The prevalence of those conditions in Latinos was 17% (95% CI: 14–20%, I^2^: 99.3%). Notably, a number of studies [16,37,39,43,52,65,70,78,80,82] exhibited greater heterogeneity compared with the others regarding population size, outcome measurement method, study design, and country of origin of immigrant Latinos.

In the further analyses, a meta-analysis of the prevalence of type 2 diabetes mellitus or high blood glucose was conducted on seven studies for immigrant Latinos and five for US-born Latinos (Figure 5). However, population size and outcome measurement methods differed in three of the studies for immigrant Latinos [39,43,54] and three for US-born Latinos [38,43,54]. A higher prevalence of the conditions was found for US-born Latinos (25% (95% CI: 16–33%, I^2^: 99.6%)) compared with immigrant Latinos (19% (95% CI: 14–24%, I^2^: 99.3%)).

##### General Obesity and Abdominal Obesity for Latinos in the USA

Of the 41 studies assessing general obesity (BMI ≥ 30 kg/m^2^), 26 collected measurements of weight and height of participants [14,15,26,27,30,31,32,33,34,37,40,43,46,50,51,52,60,61,63,64,65,66,67,72,73,74] (one study classified participants as “overweight or obese” [31] on the basis of a BMI ≥ 25 kg/m^2^), 12 obtained the parameters from self-reports [16,29,35,41,44,47,54,58,70,71,76,79], while 2 studies employed both collection methods [42,78]. All eight studies assessing abdominal obesity performed waist circumference measurements.

For general obesity, the 28 studies included in the meta-analysis (Figure 6) were heterogeneous, revealing a general obesity rate of 37% (95% CI: 34–40%, I^2^: 99.6%) in Latinos. A total of eleven [16,29,34,35,37,43,54,65,66,70,74] studies showed greater heterogeneity, particularly for population sample size, age, country of origin, and outcome measures.

For abdominal obesity, the prevalence for Latinos determined by the meta-analysis of five studies was 54% (95% CI: 48–59%, I^2^: 96.7%). Notably, all five studies were heterogeneous (Figure 7).

In our analyses of general obesity rates in immigrant Latinos and US-born Latinos, 13 studies were included in the meta-analysis for immigrants and 10 for US-born Latinos (Figure 8). We observed a higher prevalence of general obesity in immigrants (23%, 95% CI: 19–26%, I^2^: 99.7%) compared with US-born Latinos (15%, 95% CI: 13–16%, I^2^: 99%). All articles included in the meta-analysis involving immigrants, as well as those of US-born immigrants, were heterogeneous. However, five studies of immigrant Latinos [30,37,43,51,72] and one of US-born Latinos [72] exhibited greater heterogeneity for population size, age, and country of origin.

##### HDL Cholesterol and Triglycerides for Latinos in the USA

All studies that determined HDL cholesterol HDL (7) and triglycerides (7) collected data through blood workups. Five studies were included in the meta-analysis of the prevalence of low HDL cholesterol in Latinos (Figure 9), revealing a prevalence of 42% (95% CI: 35–49%, I^2^: 92.2%). The studies were heterogeneous, maybe due to different sample sizes.

For high triglycerides, the five studies included in the meta-analysis were also heterogeneous (Figure 10). Two of the studies [34,65] differed more for population size and country of origin of immigrant Latinos. The prevalence of high triglycerides in Latinos was 38% (95% CI: 24–51%, I^2^: 98.6%).

##### Metabolic Syndrome for Latinos in the USA

Of the six studies assessing the MetS, five used objective measures for obtaining data [40,60,64,65,77], while one collected data using self-reports [16].

Despite the greater heterogeneity of two studies [40,65], attributed to population size and country of origin of immigrant Latinos, they were included, together with three other studies in the meta-analysis of the prevalence rate of MetS in Latinos (Figure 11), which results showed a prevalence of 39% (95% CI: 32–45%, I^2^: 93.2%).

#### 3.4.2. Further Sensitivity Analyses

Sensitivity analyses considering sample size, extreme estimates, and study design did not show substantial changes in the results of the meta-analyses for almost all analyses of the outcomes hypertension or high blood pressure, type 2 diabetes mellitus or high blood glucose, and overall obesity. However, a statistically significant difference was observed between groups (data not shown) when considering study design (longitudinal and cross-sectional) for type 2 diabetes mellitus or high blood glucose (*p* = 0.02, prevalence of 34% for longitudinal and 15% for cross-sectional) and sample size (≤1000 participants and >1000 participants) for general obesity (*p* = 0.007, prevalence of 45% for ≤1000 and 32% for >1000). The difference observed for the study design may have occurred because longitudinal studies have greater methodological rigor and control for potential confounders than cross-sectional ones.

Further sensitivity analyses including sex, age, country of birth, length of residence in the USA, migration status, occupation, and health insurance, which could explain other differences, could not be conducted because insufficient studies had the necessary data for the estimates.

#### 3.4.3. Quality of Evidence

After applying the GRADE system criteria (Table 7), the categories risk of bias (heterogeneity) and imprecision were downgraded by one point for the outcomes arterial hypertension, type 2 diabetes mellitus, general obesity, and abdominal obesity. Publication bias was not downgraded, not even for indirect evidence, since the surrogate outcomes observed (high blood pressure and glucose, and BMI and waist measures) were strongly associated with the outcomes of interest.

The quality of evidence for the assessment of high triglycerides and low HDL cholesterol was downgraded by one point for risk of bias, inconsistency (heterogeneity), and publication bias, and by two points for imprecision, because we found only a few studies assessing those outcomes. Most studies showed major differences in sample size and broad confidence intervals in the meta-analysis. No downgrades for indirect evidence were made since no surrogate outcomes were identified.

For MetS, the risk of bias and publication bias categories were each downgraded by one level, whereas the inconsistency (heterogeneity) and imprecision categories were each downgraded by two levels. The quality of evidence was downgraded for publication bias because few studies were found assessing the MetS in immigrant Latinos, which may have led to overestimation of the measure.

The studies included in the review were expected to be heterogeneous, chiefly owing to cultural disparities, such as dietary habits and different lifestyles, among immigrant populations. Those differences can impact the outcomes observed, confirmed by I^2^ results exceeding 90%. Other factors, such as socioeconomic differences, migration status, education, acculturation, and length of residence in the USA, can also affect heterogeneity. Those factors, however, could not be explored further in our review because few studies reported those characteristics. The downgrading of quality by one level due to risk of bias for all outcomes observed was based on the individual assessment of risk of bias by the JBI.

In conclusion, the quality of evidence for the outcomes of interest of this systematic review and meta-analysis was defined as low for systemic arterial hypertension, type 2 diabetes mellitus, general obesity, and abdominal obesity and as very low for high triglycerides, low HDL cholesterol, and MetS. It is likely that further research may impact this conclusion and change it.

## 4. Discussion

### 4.1. Summary of Evidence

A total of 60 sixty studies were included in this systematic review, of which 52 were included in the meta-analysis. The overall population of the 52 studies was 436,654 immigrant Latinos in the US. The pooled prevalence obtained for arterial hypertension, type 2 diabetes mellitus, general obesity, and abdominal obesity were 28% (95% CI: 23–33%), 17% (95% CI: 14–20%), 37% (95% CI: 33–40%) and 54% (95% CI: 48–59%), respectively (Table 7). The prevalence for the other outcomes are not shown because of uncertainty regarding those outcomes in immigrant Latinos and the very low quality of the evidence.

Higher prevalence for the outcomes arterial hypertension or high blood pressure and type 2 diabetes mellitus or high blood glucose were found in US-born Latinos compared with immigrant Latinos. By contrast, the prevalence for general obesity was higher in immigrants. Data from the USA Centers for Disease Control and Prevention (CDC) [12] show an alarming rise in obesity, hypertension, and diabetes in the Latino population. The evidence found in this review demonstrates an increased cardiovascular risk in Latino populations in the USA and supports the notion of increased weight and chronic conditions in this group as a result of the accelerated process of nutrition transition occurring in many countries of Latin America and other developing countries. Secondly, the evidence suggests that exposure to a highly obesogenic environment such as the USA should be taken into account in studies on obesity and associated factors [37,51]. Thirdly, the evidence supports the potential association of socioeconomic status, variations in physical activity behavior, and significant cultural variations among Latin American countries with the development of chronic diseases.

Comparison of the prevalence found in our meta-analysis with those of the CDC for the Latino population in the six states with the largest concentration of Latino and Brazilians (Florida, Massachusetts, New Jersey, New York, California, and Connecticut) [12], based on 2019 American Community Survey (ACS) data [83], revealed higher prevalence in our meta-analysis for hypertension (CDC prevalence range: 22.4–27.4%), diabetes (CDC range: 8.3–13.9%), and high cholesterol (CDC range: 27.3–31.8% based on meta-analysis rates for triglycerides and HDL cholesterol). The meta-analysis prevalence rate for obesity was similar to the 2020 CDC prevalence [12] for general obesity (range: 27.1–41.4%) but lower only for the rate observed in California.

### 4.2. Comparison with Previous Systematic Reviews and Meta-Analyses

To the best of our knowledge, there have been no previous systematic reviews or meta-analyses assessing factors associated with MetS in immigrant Latinos in the USA, precluding any meaningful comparisons. However, two systematic reviews assessing obesity alone [17,18] in immigrant populations were conducted. Those reviews suggested a potential relationship between increased obesity and higher level of acculturation, sex, nativity, length of residence in the USA, and generational status of immigrants. Nevertheless, the prevalence of obesity in this population was not addressed in those reviews, preventing any comparisons with obesity data we found in our review and meta-analysis.

The review by Delavari et al. [17] explored the relationship between acculturation and overweight/obesity in adult immigrants from low- and middle-income countries. Overall, of the nine studies reviewed, seven identified a positive association between level of acculturation and bodyweight variables. Among the studies reviewed by those authors involving immigrant Latino populations (Mexican-Americans) in the USA, significant BMI differences were found according to acculturation levels, sex, nativity, length of residence in the country, and generational status of immigrants. That review [17] showed that higher levels of acculturation were associated with greater BMI in immigrant Latinos, and also identified an association of sex, immigrant generational status, and nativity with increased obesity.

In the analyses stratified by sex, women immigrants residing in the USA for over 15 years had a mean BMI that was 2.38 kg/m^2^ greater than the average found for women with less than 5 years of residence. A similar result was found for men living in the USA for over 15 years compared with those residing in the country for less than 5 years (1.10 kg/m^2^ higher mean BMI in former group). To assess the risk of developing obesity associated with increased level of acculturation, the same review found a 4% and 3% higher risk of obesity in immigrant men and women with a high level of acculturation, respectively. With regard to the generational status of immigrants, Delavari et al. [17] found that second and third generations of Mexican-Americans had higher BMIs than their first generation counterparts.

The review by Oza-Frank and Cunningham [18] investigated the relationship between length of residence of immigrants in the USA and BMI. For the majority of the studies reviewed, the authors identified a positive association between time of residence in the USA and BMI. Some of the study results showed that a length of residence of up to 10 years promoted no major change in the BMI of immigrants, whereas others identified peak BMI growth after 21 years of residence for men and 15 years for women. The same review [18] included four studies involving immigrant Latinos, all of which reported a significant association between BMI and length of residence in the US.

### 4.3. Limitations

Our systematic review and meta-analysis have some limitations. First, there is high statistical heterogeneity for all outcomes. Immigrant populations differ in many ways, e.g., country of origin, migration status, degree of acculturation, and educational and socioeconomic levels, and those differences might explain the high inconsistencies found. Thus, the pooled estimates in this study should be regarded as suggestive as opposed to conclusive.

Second, most studies reviewed specifying the origin of the immigrant Latino population included only immigrants from Mexico or Central America. As a result, much of the evidence found in our review is limited to a subpopulation of immigrants from Latin America. There is scant scientific evidence on components of the MetS in immigrants from South America. Thus, the evidence found in this review and meta-analysis may not be representative of those countries which exhibit major sociocultural and economic differences that can impact the development of chronic diseases.

Third, it was not possible to stratify analyses for other key characteristics, such as sex, age, country of birth, length of residence in the USA, migration status, occupation, and health insurance, because not all studies reported the detailed information needed to calculate such estimates.

Fourth, some studies reported results for more than one outcome of interest. However, it was not possible to extract information for all outcomes in some cases, as not all studies provided the data needed for that.

Fifth, some studies used measures derived from self-reports, particularly for general obesity, which are known to be subject to information bias, typically leading to an underestimation of own weight by women and overestimation by men [84].

Lastly, the studies were pooled in the meta-analyses, irrespective of risk of individual bias associated with them in order to prevent selection bias in meta-analyses due to stratification by quality of study [85].

## 5. Conclusions

This systematic review and meta-analysis were conducted in collaboration with researchers from the USA and Brazil to compile the available evidence on MetS and its risk factors in the immigrant Latino population. A large body of evidence was identified encompassing numerous studies, particularly pertaining to hypertension, type 2 diabetes, and both general and abdominal obesity. However, the evidence was classified as low or very low depending on the outcome. This rating suggests that future reviews and meta-analyses may reach different conclusions when more studies assessing MetS and its components become available, particularly HDL cholesterol and triglycerides, factors which featured less in the studies reviewed.

The pooled prevalence obtained were 28% (95% CI: 23–33%) for arterial hypertension, 17% (95% CI: 14–20%) for type 2 diabetes mellitus, 37% (95% CI: 33–40%) for general obesity, and 54% (95% CI: 48–59%) for abdominal obesity. Greater prevalence of arterial hypertension and type 2 diabetes mellitus were found for US-born Latinos, while general obesity rates were higher for immigrant Latinos.

The production of prevalence estimates of the MetS and its risk factors in the immigrant population appears to be justified and evidence-based, considering the cultural, dietary, and lifestyle differences encountered by migrants, which promote increased obesity (strongly associated with MetS) and chronic diseases associated with the condition.

However, few studies were available that included, or analyzed separately, information on immigrant Latinos from South America, including Brazil, a region for which only two studies were found. Therefore, further studies are needed addressing MetS and its factors in immigrants from South America, particularly Brazil, in view of the particularities regarding the culture, language, and diet compared with other Latin American nations.

## Figures and Tables

**Figure 1 ijerph-20-01307-f001:**
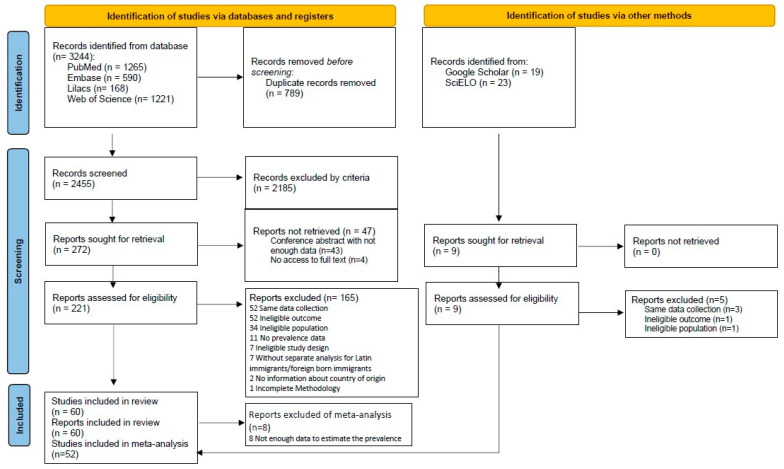
Flow diagram of study selection process following the PRISMA 2020 flow diagram for new systematic reviews which included searches of databases, registers and other sources. Source: model extracted from Page et al. [21].

**Figure 2 ijerph-20-01307-f002:**
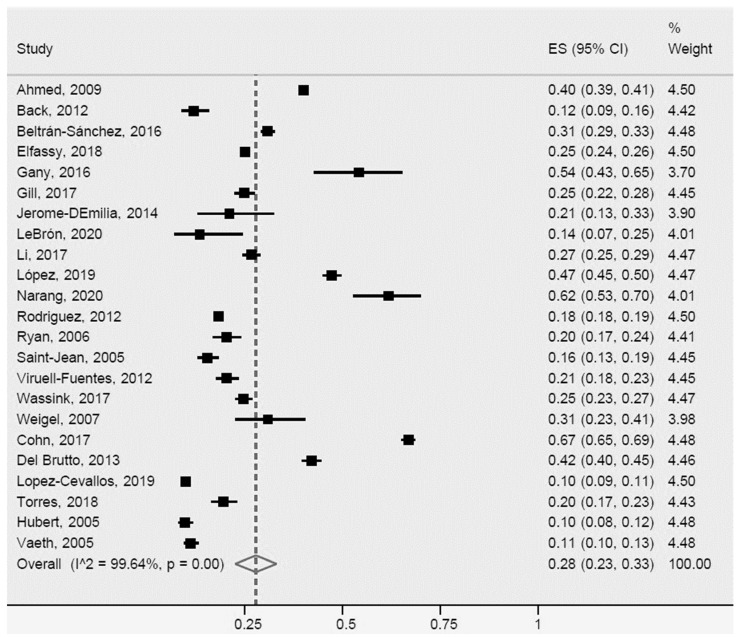
Arterial hypertension or high blood pressure for Latinos in the US. ES = Estimated proportion/prevalence; CI = Confidence interval; I^2 = I^2^ index.

**Figure 3 ijerph-20-01307-f003:**
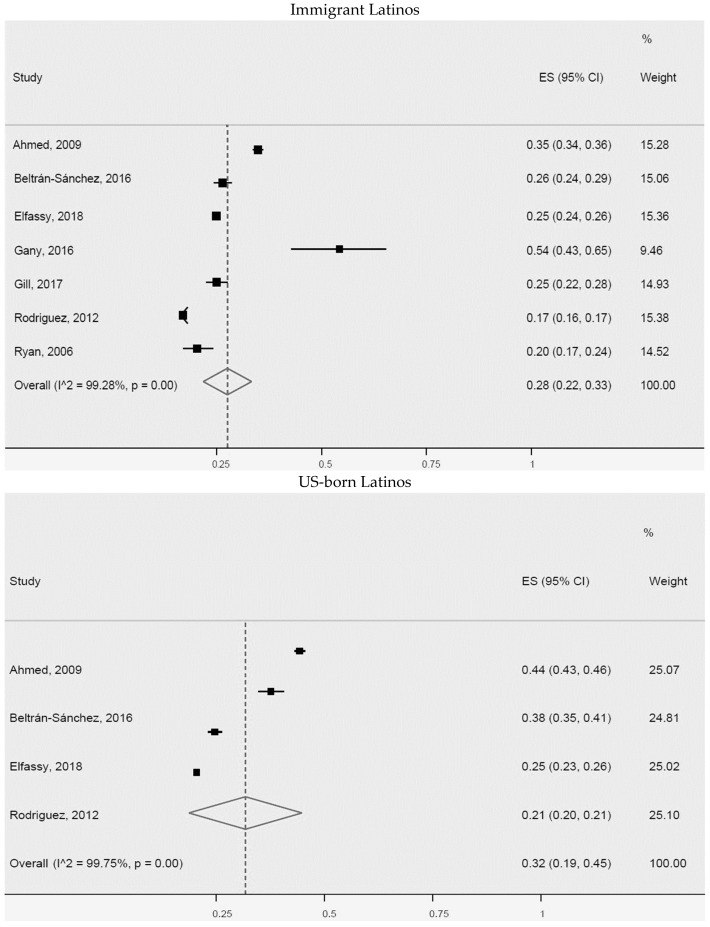
Arterial hypertension or high blood pressure for immigrant Latinos and US-born Latinos. ES = Estimated proportion/prevalence; CI = Confidence interval; I^2 = I^2^ index.

**Figure 4 ijerph-20-01307-f004:**
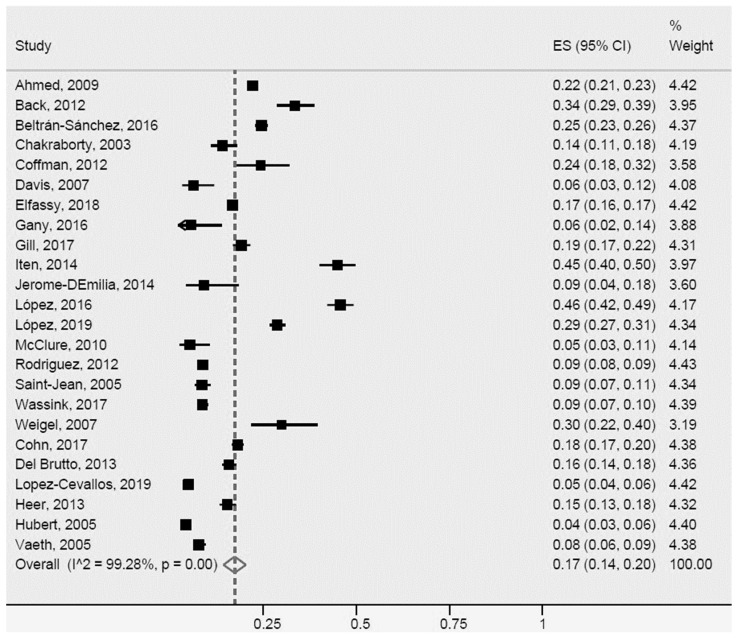
Type 2 diabetes mellitus or high blood glucose for Latinos in the US. ES = Estimated proportion/prevalence; CI = Confidence interval; I^2 = I^2^ index.

**Figure 5 ijerph-20-01307-f005:**
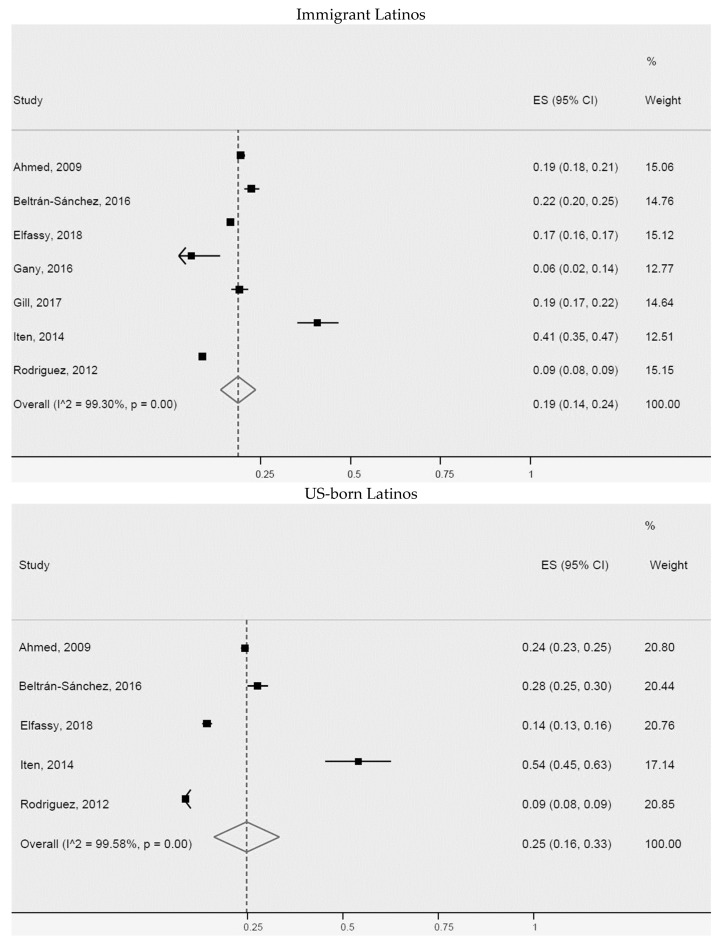
Type 2 diabetes mellitus or high blood glucose for immigrant Latinos and US-born Latinos. ES = Estimated proportion/prevalence; CI = Confidence interval; I^2 = I^2^ index.

**Figure 6 ijerph-20-01307-f006:**
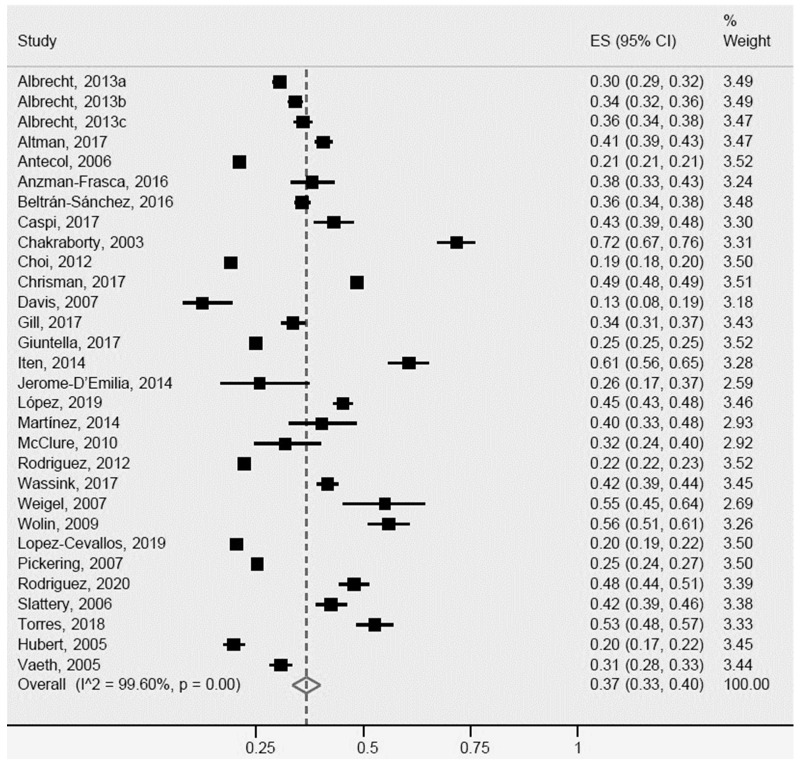
General obesity for Latinos in the US. ES = Estimated proportion/prevalence; CI = Confidence interval; I^2 = I^2^ index.

**Figure 7 ijerph-20-01307-f007:**
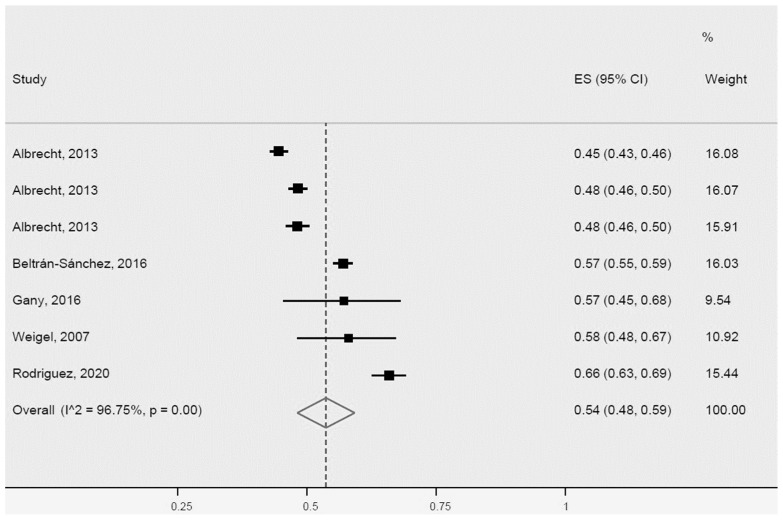
Abdominal obesity for Latinos in the USA. ES = Estimated proportion/prevalence; CI = Confidence interval; I^2 = I^2^ index.

**Figure 8 ijerph-20-01307-f008:**
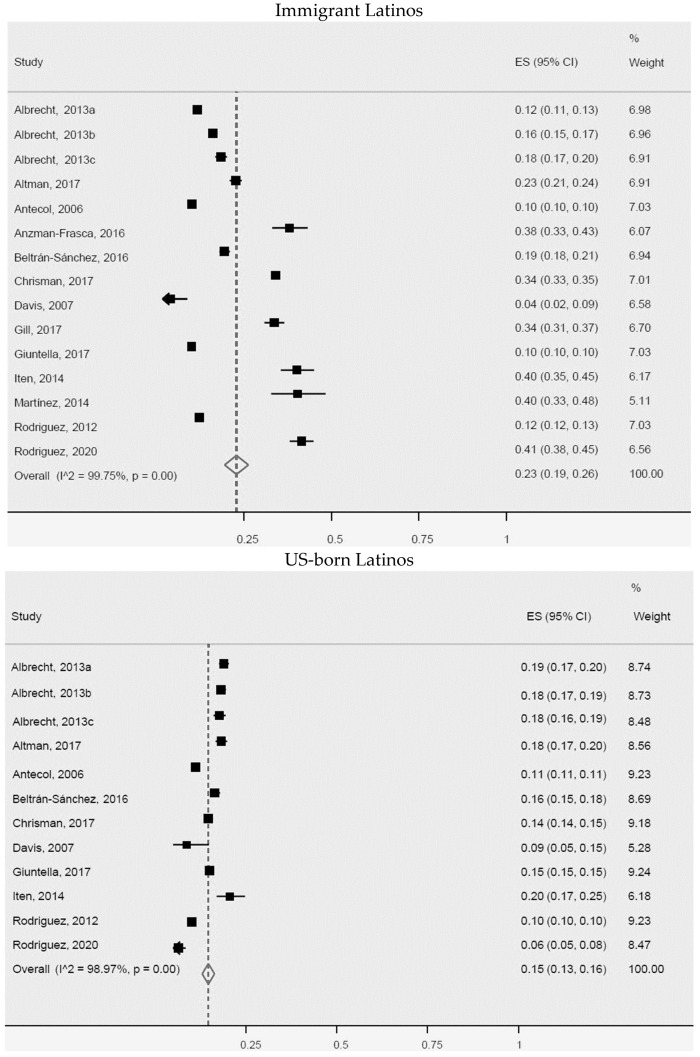
General obesity for immigrant Latinos and US-born Latinos. ES = Estimated proportion/prevalence; CI = Confidence interval; I^2 = I^2^ index.

**Figure 9 ijerph-20-01307-f009:**
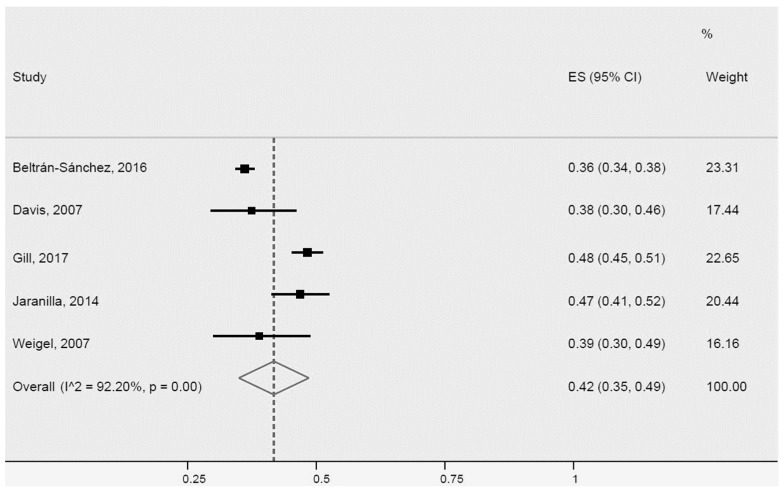
Low HDL cholesterol for Latinos in the USA. ES = Estimated proportion/prevalence; CI = Confidence interval; I^2 = I^2^ index.

**Figure 10 ijerph-20-01307-f010:**
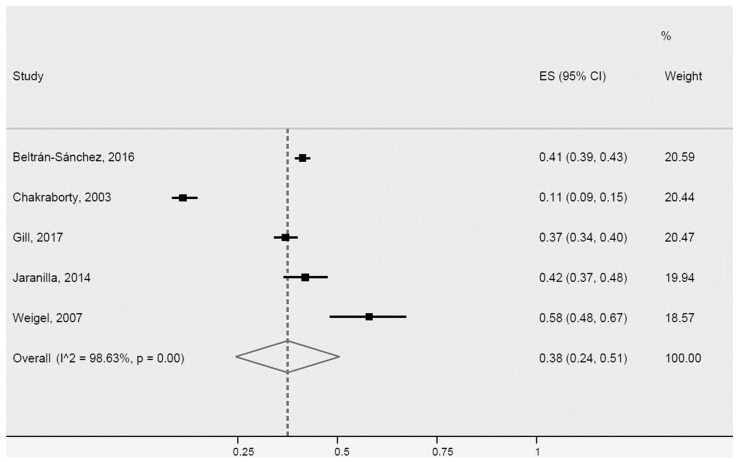
High triglycerides for Latinos in the USA. ES = Estimated proportion/prevalence; CI = Confidence interval; I^2 = I^2^ index.

**Figure 11 ijerph-20-01307-f011:**
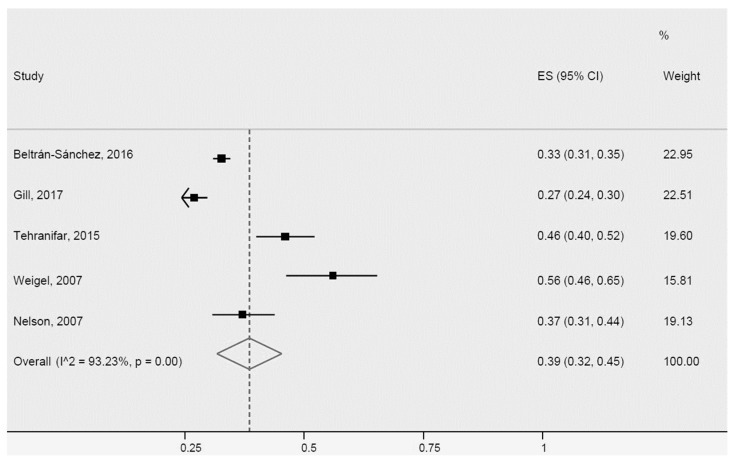
Metabolic Syndrome for Latinos in the USA. ES = Estimated proportion/prevalence; CI = Confidence interval; I^2 = I^2^ index.

**Table 1 ijerph-20-01307-t001:** Exposures Assessed in Studies Reviewed.

Immigration-Related	Immigrant-Related	Health-, Dietary-, and Lifestyle-Related	Community-Related
generational status [14]	nativity [26,35,42,54,72]	health behaviors [16,34]	immigrant concentration [49,68]
length of residence in USA [14,26,36,39,47,72,81]	agricultural work [31,33,61]	health assimilation; health literacy; nutrition transition; dietary characteristics [41,47,51,80]	community factors [58,68]
migration history [34]	income; sociodemographic factors [31,32,47,53,68,70]	healthcare services access and utilization [56,62]	
immigration [15,27,28,29,30,31,45,82]	ethnicity [37,44]	food insecurity [64,65]	
acculturation [16,50,54,58,66,73,78,79,80]	occupation [44]	lifestyle; environment; lifestyle predictors; physical activity patterns [53,69,73]	
age at immigration [79]	age [81]	hypertension; type 2 diabetes; MetS; family history of type 2 diabetes; cardiovascular risk factors [57,60,68,75,77]	
migrant status [43,44,81]	social determinants of health [67]	psychiatric disorders; substance use [71]	
discrimination [48,52,55]		worry about deportation [74]	
residential mobility [42]		health status [16]	

MetS = Metabolic Syndrome.

**Table 2 ijerph-20-01307-t002:** Characteristics of Latinos in the USA studies reviewed.

Study ID	Total Study Participants	Female	Male	Age Group	Country of Birth	Comparison Group	Time in the USA	Study Design	Study Period	Exposures	Immigrant Worker	Occupation	Documentation	Insurance
Ahmed, 2009 [14]	70,110	0	70,110	45–69		US-born black and white	<10 y; 11–15 y; 16–25 y; >25 y	cohort	2002–2003	generational status; length of residency in the US	No			
Albrecht, 2013 [26]	8149	3914	4235	20–64	Mexico	US-born Mexicans	<10 y; >10 y	cross-sectional	1988–1994; 1999–2008	nativity; length of residency in the US	No			
Altman, 2017 [27]	25,499	25,499	0	20–64	Mexico	Mexicans in Mexico; US-born Mexicans; US-born non-Hispanics whites/blacks		cross-sectional	1999–2009 NHANES (US); 2006 ENSANUT (Mexico)	immigration	No			
Angel, 2008 [28]	1975	1153	822	20—not specified		non-Hispanic whites, blacks, and Asians	<10 y; >10 y	cross-sectional	June–December 2004	immigration	No			Yes
Antecol, 2006 [29]	490,806	258,718	232,088	20–64		natives (Hispanics, whites, and blacks)	0–4 y; 5–9 y; 10–14 y; 15 y+	cross-sectional	1989–1996	immigration	No			
Anzman-Frasca, 2016 [30]	345	345	0	20–55	Brazil; Haiti; El Salvador; Colombia; Guatemala; Dominican Republic; Honduras	between groups	<10 y	cross-sectional	March and June 2010 (first wave); May and June 2011 (second wave)	immigration	No			
Back, 2012 [82]	1096	441	655	30—not specified	Guyana; other			case–control	1 May 2004–30 April 2006	immigration	No			
Beltrán-Sánchez, 2016 [15]	15,957	9022	6935	20–50+	Mexico	US-born Mexican-American; Mexican living in Mexico; non-Hispanic whites		cross-sectional	ENSANUT 2006; NHANES 1999–2000 and 2009–2010	immigration	No			
Boggess, 2016 [31]	793,188	427,528	365,660	18–60+				cross-sectional	2012	agricultural work (seasonal and migratory); immigration; low income	Yes	agricultural workers (seasonal or migratory)		Yes
Briones, 2016 [76]	31,305	11,402	19,903	<35–65+	Mexico			cross-sectional	2010		No		Yes	
Caspi, 2017 [32]	800	636	164	18–60+	Puerto Rico; Haiti; other	Us-born	0–5 y; 5–10 y; 10 y+	cross-sectional	February 2007 and June 2009	low-income immigrants	No			
Castañeda, 2015 [33]	282	163	118			migrant or seasonal status		cross-sectional	2002–2004	migrant or seasonal status	Yes	farmworkers		Yes
Chakraborty, 2003 [34]	390	390	0	18–65	Mexico; other		<5 y; 5–9 y; 10 y+	cross-sectional	1993	migration history; health behavior changes—mediator	No			
Choi, 2012 [35]	7786	4287	3499		Central and South America	origin	<1 y; 1–4 y; 5–9 y; 10 y+	cross-sectional	2003	nativity	No			
Chrisman, 2017 [79]	18,298	14,048	4250	21–94	Mexico	US-born		cohort	Start year 2001	language acculturation; age at immigration	No			
Coffman, 2012 [36]	144	113	31		Mexico; Central and South America; other			cross-sectional		recent Latino immigrants	No			Yes
Cohn, 2017 [68]	3317	1523	1794	30–74		mon-Hispanic whites		cross-sectional	2010 (US Census); 2012–2013 (hospital data)	CVD risk factors (individual factors); median household income and Hispanic concentration (neighborhood-level)	Yes			Yes
Davis, 2007 [37]	189	98	91	18–40	Central America	African-American; US-born African Caribbean		cross-sectional		ethnicity	No			Yes
Dawson, 2019 [67]	181	120	61	18–64	Mexico; Central America		Mean 21.6 (16.2); Median 19	cross-sectional		social determinants of health	Yes			
Del Brutto, 2013 [69]	3850	2413	1437	40—not specified	Dominic Republic; Puerto Rico; Cuba; other	coastal Ecuador population (Atahualpa)		cross-sectional	1993–2001; June–October 2012	lifestyle; environment	No			
Elfassy, 2018 [38]	16,156	8498	7658		Central and South America	between groups	<10 y; >10 y	cross-sectional	2008–2011		No			
Gany, 2016 [39]	413	1	412	18–60+	Central and South America		<10 y; >10 y; US-born	cross-sectional	September–October 2011	length of residence in the US	Yes	taxi drivers		Yes
Gill, 2017 [40]	1042	704	338	18–87	El Salvador; Honduras; Peru; Guatemala; Bolivia	between groups	Mean: 8.8 years	cross-sectional			No			Yes
Giuntella, 2017 [41]	729,793	387,502	342,291	25–64		US born	<10 y; 10–15 y; 15–20 y; >20 y	cross-sectional	1989–2014	health assimilation	Yes			
Glick, 2015 [42]	525	372	153	18–60	Mexico	Mexicans in Mexico	<10 y; 10 y+ speaking Spanish; 10 y+ English or bilingual; US-born	cross-sectional	March 2009 (US); May–June 2009 (Mexico)	nativity; residential mobility	No			
Heer, 2013 [75]	1002	660	342		Mexico	Mexican-Americans without diabetes	Diabetes: mean 40.9 y (SD 18.6) No diabetes: mean 33.0 y (SD 17.5)	cross-sectional	November 2009–May 2010	diabetes	No			Yes
Hubert, 2005 [16]	1005	380	521	18–64	Mexico; other		Mean	cross-sectional	July–December 2000	health status; health behaviors; acculturation	Yes	skilled professional; semiskilled white-collar, clerical; semiskilled blue-collar; unskilled service, laborer; farmworker; homemaker; unemployed or student		
Iten, 2014 [43]	401	207	194		Mexico	US-born Mexican-Americans; documented Mexican immigrants		cross-sectional	2008–2009	immigrant status	Yes		Yes	
Jackson, 2014 [44]	175,244	28,730	30,484	18–65+		non-Hispanic whites and blacks	<15 y; 15 y+	cross-sectional	2004–2011 (NHIS)	immigrant status; race/ethnicity occupation	Yes	professional/management; support services		
Jaranilla, 2014 [45]	59,791	33,025	26,766	20–65+	Central and South America	US born		cross-sectional	January–December 2010	immigration	No			Yes
Jerome-D’Emilia, 2014 [46]	66	66	0	21–79	Puerto Rico; Dominican Republic; Mexico; other	between groups		cross-sectional			Yes		Yes	Yes
Klabunde 2020 [47]	361	191	170	18–74	Brazil		Mean: 12.7 (SD 6.7)	cross-sectional	December 2013–March 2014	socio-demographic factors; dietary characteristics; length of residence in the US	Yes			
LeBrón, 2020 [48]	213	138	75			non-Hispanic whites and blacks		cross-sectional	2002–2003 2007–2008	discrimination	No			
Li, 2017 [49]	1563	1080	483	18–91	Puerto Rico; Mexico; other			cross-sectional	2006–2008 (Survey) 2005–2009 (ACS)	immigrant concentration; Latino density	No			Yes
Lopez-Cevallos, 2019 [70]	3382	673	2709	18–74			0–5 y; 6–9 y; >10 y	cross-sectional	2004–2012	sociodemographic factors	Yes	farmworkers		
López, 2016 [80]	744	405	339	30–72	Puerto Rico; Dominican Republic; other		5–10 y; 10–15 y; 15–20 y; >20 y	cohort	January 2010–March 2012	acculturation; health literacy	No			Yes
López, 2019 [50]	1818	1187	631	45—not specified			<5 y; 5–10 y; 11–20 y; >20 y	cross-sectional		acculturation	No			
Martínez, 2014 [51]	149	98	51	20–77	Mexico; Central and South America		Mean: 10.24 (SD 10.12)	cross-sectional	2011	nutrition transition	No			
McClure, 2010 [52]	132	86	46		Mexico	US population	Women: 9.5 (SD 6.9); Men: 13.5 (SD 9.4)	cross-sectional		discrimination	No			
Narang, 2020 [53]	983	0	983	19–76			≤2 y; 3–9 y; 10–15 y; ≥16 y	cross-sectional	December 2010–November 2017	demographic factors; lifestyle predictors	Yes	taxi drivers; for-hire vehicle drivers		
Nelson, 2007 [77]	205				Mexico		Mean 25.7 (SD 16.4)	cross-sectional	April 2004–October 2005	family history of diabetes	No			
Pickering, 2007 [71]	43,093	24,575	18,518	18–65+				cross-sectional	2001–2002	psychiatric disorders (mood, anxiety, and personality disorders); substance use (alcohol, drugs, and nicotine)	No			
Rodriguez, 2012 [54]	160,081	81,164	78,917		Mexico; Central and South America; other	US-born Hispanics; non-Hispanic white	≤1–4 y; 5–9 y; 10–14 y; >15 y	cross-sectional	2001, 2003, 2005, and 2007	acculturation; nativity	No			Yes
Rodriguez, 2020 [72]	787	787	0	40–65	Dominican Republic; Puerto Rico; Cuba; Mexico	US-born		cross-sectional	2012–2018	nativity; migration timing	No			
Ryan, 2006 [55]	666	453	213			African-Americans	Latino Immigrants: mean 4.47	cross-sectional	2002–2003	discrimination	Yes			Yes
Saint-Jean, 2005 [56]	680	340	340	0–75+	Haiti	with and without insurance	<5 y; 5–10 y; 11–14 y; 15 y+	cross-sectional	2001	health services utilization	No			Yes
Salinas, 2014 [81]	1936	1302	634	18–80	Mexico	US born	≤10 y; >10 y; US-born	cohort	2004–2007	immigrant status; length of residence in the US; age	No			
Shelley, 2011 [57]	2585	1592	993			non-Hispanic whites and blacks		cross-sectional	2007–2008	hypertension	No			Yes
Shi, 2015 [58]	15,471	8049	7422	18–65+		White	0–4 y; 5–10 y; 10 y+; US-born	cross-sectional	2005 and 2007	acculturation; community factors	No			
Singh-Franco, 2013 [59]	114	85	29			non-Haitians		cross-sectional	January 2003–May 2008	intervention of a multidisciplinary team	No			
Slattery, 2006 [73]	2039	2039	0	<40–79		non-Hispanic whites		cross-sectional		language acculturation; physical activity patterns	No			
Tehranifar, 2015 [60]	373	373	0	40–64		with and without MetS		cross-sectional	November 2012 and May 2014	MetS	No			
Torres, 2018 [74]	545	545	0		Mexico		<15 y; 16–20 y; >21 y; USA born	cross-sectional	March 2012 August 2014	worry about deportation	No			
Vaeth, 2005 [78]	1163	624	539	18–65	Mexico; El Salvador; Guatemala; Honduras		<5 y; 5–10 y; 10 y+	cross-sectional	July 2000–October 2002	acculturation	No			Yes
Villarejo, 2010 [61]	654	238	416		Mexico; other	males and females	Males: median 14 Females: median 9	cross-sectional	1999	agricultural work	Yes	farmworkers	Yes	
Viruell-Fuentes, 2012 [62]	804	456	348					cross-sectional	May 2001–March 2003	access to care; neighborhood effects	No			
Wassink, 2017 [63]	3731	1921	1810	24–32	Mexico; Cuba; Central and South America	migrant generation, blacks		cross-sectional	Phase I (1994–1995), III (2001–2002) and IV (2008–2009).		No			Yes
Weigel, 2019 [64]	75	67	8	40–84	Mexico	food insecure; food secure	Mean: 19.9 ± 15	cross-sectional	April-May 2015	food insecurity	Yes			
Weigel, 2007 [65]	100	43	57	18–61+	Mexico		≤10 y	cross-sectional	10-month period in 2003	food insecurity	No			
Wolin, 2009 [66]	388	388	0	40–77	Mexico; other		≤10 y; 11–20 y; >20 y	cross-sectional	November 2000 and June 2002 (Phase I); May 2003 and June 2004 (Phase II)	acculturation	Yes	homemaker; other		

**Table 3 ijerph-20-01307-t003:** JBI Critical Appraisal Checklist for Analytical Cross-Sectional Studies.

	**Albrecht 2013**	**Altman 2017**	**Angel 2008**	**Antecol 2006**	**Anzman-Frasca 2016**	**Beltrán-Sánchez 2016**	**Boggess 2016**	**Briones 2016**	**Caspi 2017**	**Castañeda 2015**	**Chakraborty 2003**	**Choi 2012**	**Coffman 2012**	**Cohn 2017**	**Davis 2007**	
1. Were the criteria for inclusion in the sample clearly defined?	Y	Y	Y	Y	Y	Y	Y	U	Y	Y	Y	Y	Y	Y	Y	
2. Were the study subjects and the setting described in detail?	Y	Y	Y	Y	Y	Y	Y	Y	Y	Y	Y	Y	Y	N	Y	
3. Was the exposure measured in a valid and reliable way?	Y	Y	Y	Y	Y	Y	Y	U	Y	Y	Y	Y	Y	Y	Y	
4. Were objective, standard criteria used for measurement of the condition?	Y	Y	Y	Y	Y	Y	Y	U	Y	Y	Y	Y	Y	Y	Y	
5. Were confounding factors identified?	Y	Y	Y	Y	Y	Y	N	N	Y	Y	Y	U	U	U	N	
6. Were strategies to deal with confounding factors stated?	Y	Y	Y	Y	Y	Y	N	N	Y	Y	Y	U	U	U	N	
7. Were the outcomes measured in a valid and reliable way?	Y	Y	Y	Y	Y	Y	Y	U	Y	Y	Y	Y	Y	Y	Y	
8. Was appropriate statistical analysis used?	Y	Y	Y	Y	Y	Y	Y	Y	Y	Y	Y	Y	Y	Y	Y	
% of “yes”	100%	100%	100%	100%	100%	100%	75%	25%	100%	100%	100%	75%	75%	62.5%	75%	
Risk of bias	low	low	Low	low	low	low	low	high	low	low	low	low	low	moderate	low	
	**Dawson 2019**	**Del Brutto 2013**	**Elfassy 2018**	**Gany 2016**	**Gill 2017**	**Giuntella 2017**	**Glick 2015**	**Heer 2013**	**Hubert 2005**	**Iten 2014**	**Jackson 2014**	**Jaranilla 2014**	**Jerome-D’Emilia 2014**	**Klabunde 2020**	**LeBrón 2020**	**Li 2017**
1. Were the criteria for inclusion in the sample clearly defined?	Y	Y	Y	N	Y	Y	Y	Y	Y	Y	Y	Y	Y	Y	Y	Y
2. Were the study subjects and the setting described in detail?	Y	Y	Y	N	Y	Y	Y	Y	Y	Y	Y	Y	Y	Y	Y	Y
3. Was the exposure measured in a valid and reliable way?	Y	Y	Y	Y	Y	Y	Y	Y	Y	Y	Y	Y	U	Y	Y	Y
4. Were objective, standard criteria used for measurement of the condition?	Y	Y	Y	Y	Y	NA	Y	Y	Y	Y	Y	Y	U	Y	Y	Y
5. Were confounding factors identified?	U	Y	Y	Y	Y	Y	Y	Y	U	Y	Y	Y	N	Y	Y	Y
6. Were strategies to deal with confounding factors stated?	U	Y	Y	Y	Y	Y	Y	Y	U	Y	Y	Y	N	Y	Y	Y
7. Were the outcomes measured in a valid and reliable way?	Y	Y	Y	Y	Y	Y	Y	Y	Y	Y	Y	Y	U	Y	Y	Y
8. Was appropriate statistical analysis used?	Y	Y	Y	Y	Y	Y	Y	Y	Y	Y	Y	Y	Y	Y	Y	Y
% of “yes”	75%	100%	100%	75%	100%	87.5%	100%	100%	75%	100%	100%	100%	37.5%	100%	100%	100%
Risk of bias	low	low	low	low	low	low	low	low	low	low	low	low	high	low	low	low
		**Lopez-Cevallos 2019**	**López 2019**		**Martínez 2014**	**McClure 2010**	**Narang 2020**	**Nelson 2007**	**Pickering 2007**	**Rodriguez 2012**		**Rodriguez 2020**	**Ryan 2006**	**Saint-Jean 2005**		**Shelley 2011**
1. Were the criteria for inclusion in the sample clearly defined?		Y	Y		Y	N	Y	Y	Y	Y		Y	Y	Y		Y
2. Were the study subjects and the setting described in detail?		Y	Y		Y	N	N	Y	Y	Y		Y	Y	Y		Y
3. Was the exposure measured in a valid and reliable way?		Y	Y		Y	Y	Y	Y	Y	Y		Y	Y	Y		U
4. Were objective, standard criteria used for measurement of the condition?		Y	Y		Y	Y	Y	Y	Y	Y		Y	Y	Y		Y
5. Were confounding factors identified?		Y	Y		Y	Y	Y	Y	Y	Y		Y	Y	U		U
6. Were strategies to deal with confounding factors stated?		Y	Y		Y	Y	Y	Y	Y	Y		Y	Y	U		U
7. Were the outcomes measured in a valid and reliable way?		Y	Y		Y	Y	Y	Y	Y	Y		Y	Y	Y		Y
8. Was appropriate statistical analysis used?		Y	Y		N	Y	Y	Y	Y	Y		Y	Y	Y		Y
% of “yes”		100%	100%		87.5%	75%	87.5%	100%	100%	100%		100%	100%	75%		62.5%
Risk of bias		low	low		low	low	low	low	low	low		low	low	low		moderate
		**Shi 2015**	**Singh-Franco 2013**		**Slattery 2006**	**Tehranifar 2015**	**Torres 2018**	**Villarejo 2010**	**Vaeth 2005**	**Viruell-Fuentes 2012**		**Wassink 2017**	**Weigel 2019**	**Weigel 2007**		**Wolin 2009**
1. Were the criteria for inclusion in the sample clearly defined?		Y	Y		Y	Y	Y	Y	Y	Y		Y	Y	Y		Y
2. Were the study subjects and the setting described in detail?		Y	Y		Y	Y	Y	Y	Y	Y		Y	Y	Y		Y
3. Was the exposure measured in a valid and reliable way?		Y	Y		Y	Y	Y	Y	Y	U		Y	Y	Y		Y
4. Were objective, standard criteria used for measurement of the condition?		Y	Y		Y	Y	Y	Y	Y	Y		Y	Y	Y		Y
5. Were confounding factors identified?		Y	Y		Y	Y	Y	Y	Y	U		Y	Y	Y		Y
6. Were strategies to deal with confounding factors stated?		Y	Y		Y	Y	Y	Y	Y	U		Y	Y	Y		Y
7. Were the outcomes measured in a valid and reliable way?		Y	Y		Y	Y	Y	Y	Y	U		Y	Y	Y		Y
8. Was appropriate statistical analysis used?		Y	Y		Y	Y	Y	Y	Y	Y		Y	Y	Y		Y
% of “yes”		100%	100%		100%	100%	100%	100%	100%	50%		100%	100%	100%		100%
Risk of bias		low	low		low	low	low	Low	low	moderate		low	low	low		low

Y = yes, N = no, U = unclear, NA = not applicable.

**Table 4 ijerph-20-01307-t004:** JBI Critical Appraisal Checklist for Cohort Studies.

	Ahmed 2009	Chrisman 2017	López 2016	Salinas 2014
1. Were the two groups similar and recruited from the same population?	Y	Y	Y	Y
2. Were the exposures measured similarly to assign people to both exposed and unexposed groups?	Y	N	Y	Y
3. Was the exposure measured in a valid and reliable way?	Y	Y	Y	Y
4. Were confounding factors identified?	Y	Y	Y	U
5. Were strategies to deal with confounding factors stated?	Y	Y	Y	U
6. Were the groups/participants free of the outcome at the start of the study (or at the moment of exposure)?	U	U	N	U
7. Were the outcomes measured in a valid and reliable way?	Y	Y	Y	Y
8. Was the follow-up time reported and sufficient to be long enough for outcomes to occur?	N	Y	Y	Y
9. Was follow-up complete, and if not, were the reasons to loss to follow-up described and explored?	Y	N	N	N
10. Were strategies to address incomplete follow-up utilized?	NA	N	N	N
11. Was appropriate statistical analysis used?	Y	Y	Y	Y
% of “yes”	73%	64%	73%	54.5%
Risk of bias	low	moderate	low	moderate

Y = yes, N = no, U = unclear, NA = not applicable.

**Table 5 ijerph-20-01307-t005:** JBI Critical Appraisal Checklist for Case–Control Studies.

	Back 2012
1. Were the groups comparable other than the presence of disease in cases or the absence of disease in controls?	Y
2. Were cases and controls matched appropriately?	N
3. Were the same criteria used for identification of cases and controls?	Y
4. Was exposure measured in a standard, valid, and reliable way?	Y
5. Was exposure measured in the same way for cases and controls?	Y
6. Were confounding factors identified?	Y
7. Were strategies to deal with confounding factors stated?	Y
8. Were outcomes assessed in a standard, valid, and reliable way for cases and controls?	Y
9. Was the exposure period of interest long enough to be meaningful?	Y
10. Was appropriate statistical analysis used?	Y
% of “yes”	90%
Risk of bias	low

Y = yes, N = no.

**Table 6 ijerph-20-01307-t006:** Outcomes Assessed for the Latinos in the USA Studies.

Study ID	Primary Outcomes	Secondary Outcomes
Ahmed, 2009 [14]	HTN; DM; obesity	
Albrecht, 2013 [26]	Obesity; AO	
Altman, 2017 [27]	Obesity	
Angel, 2008 [28]	HTN	
Antecol, 2006 [29]	Obesity	
Anzman-Frasca, 2016 [30]	Obesity	
Back, 2012 [82]	HTN; DM	
Beltrán-Sánchez, 2016 [15]	HTN; DM; obesity; AO; low HDL; high TGL; MetS	
Boggess, 2016 [31]	HTN; DM; overweight or obesity	
Briones, 2016 [76]	Obesity	
Caspi, 2017 [32]	Obesity	
Castañeda, 2015 [33]	HTN; DM; obesity	
Chakraborty, 2003 [34]	DM; obesity; high TGL	sleep duration
Choi, 2012 [35]	Obesity	
Chrisman, 2017 [79]	Obesity	
Coffman, 2012 [36]	DM	
Cohn, 2017 [68]	HTN; DM	
Davis, 2007 [37]	DM; obesity; low HDL	
Dawson, 2019 [67]	HTN; DM; obesity	
Del Brutto, 2013 [69]	HTN; DM	
Elfassy, 2018 [38]	HTN; DM	
Gany, 2016 [39]	HTN; DM; AO	
Gill, 2017 [40]	HTN; DM; obesity; low HDL; high TGL; MetS	
Giuntella, 2017 [41]	Obesity	
Glick, 2015 [42]	Obesity	
Heer, 2013 [75]	DM	
Hubert, 2005 [16]	HTN; DM; obesity; MetS (2 or 3 factors)	
Iten, 2014 [43]	DM; obesity	
Jackson, 2014 [44]	HTN; DM2; obesity	sleep duration
Jaranilla, 2014 [45]	Low HDL; high TGL	
Jerome-D’Emilia, 2014 [46]	HTN; DM2; obesity	
Klabunde, 2020 [47]	Obesity	
LeBrón, 2020 [48]	HTN	
Li, 2017 [49]	HTN	
Lopez-Cevallos, 2019 [70]	HTN; DM2; obesity	
López, 2016 [80]	DM2	
López, 2019 [50]	HTN; DM2; obesity	
Martínez, 2014 [51]	Obesity	
McClure, 2010 [52]	Obesity; DM2	
Narang, 2020 [53]	HTN	
Nelson, 2007 [77]	HTN; DM2; AO; low HDL, high TGL; MetS	
Pickering, 2007 [71]	Obesity	
Rodriguez, 2012 [54]	HTN; DM2; obesity	
Rodriguez, 2020 [72]	Obesity; AO	
Ryan, 2006 [55]	HTN	
Saint-Jean, 2005 [56]	HTN; DM2	
Salinas, 2014 [81]	HTN	
Shelley, 2011 [57]	HTN; DM2	
Shi, 2015 [58]	HTN; obesity	
Singh-Franco, 2013 [59]	HTN; DM2	
Slattery, 2006 [73]	Obesity	
Tehranifar, 2015 [60]	HTN; DM2; obesity; AO; low HDL; high TGL; MetS	
Torres, 2018 [74]	HTN; obesity	
Vaeth, 2005 [78]	HTN; DM2; obesity	
Villarejo, 2010 [61]	HTN; DM2; obesity	
Viruell-Fuentes, 2012 [62]	HTN	
Wassink, 2017 [63]	HTN; DM2; obesity	
Weigel, 2019 [64]	HTN; DM2; obesity; AO; MetS	
Weigel, 2007 [65]	HTN; DM2; obesity; AO; low HDL; high TGL; MetS	
Wolin, 2009 [66]	Obesity	

HTN = Hypertension; DM2 = Type 2 diabetes mellitus; AO = Abdominal obesity; HDL = High Density Lipoprotein; TGL = Triglycerides; MetS = Metabolic Syndrome.

**Table 7 ijerph-20-01307-t007:** Summary of Findings.

Systematic Review of the Metabolic Syndrome and Its Components in USA Immigrants				
Population: Latin American Immigrants ≥18 Years Old Settings: USA Exposure: Immigration Comparator: US-Born Population				
Outcomes	Prevalence Estimate (%) (95% CI)	No. of Latino Participants (Studies)	GRADE Evidence Level	Comments
**Hypertension**	28 (23–33)	84.047	⊕⊕⊝⊝ ^a,b,c,g,h^ low	The available evidence is sufficient to determine the prevalence, but confidence in the estimate is limited. As more information becomes available, the observed prevalence could change, and this change may be large enough to alter the conclusion.
**Type 2 Diabetes Mellitus**	17 (14–20)	83.423	⊕⊕⊝⊝ ^a,b,c,g,h^ low	The available evidence is sufficient to determine the prevalence, but confidence in the estimate is limited. As more information becomes available, the observed prevalence could change, and this change may be large enough to alter the conclusion.
**Obesity (BMI > 30 kg/m^2^)**	37 (33–40)	237.035	⊕⊕⊝⊝ ^a,b,c,g,h^ low	The available evidence is sufficient to determine the prevalence, but confidence in the estimate is limited. As more information becomes available, the observed prevalence could change, and this change may be large enough to alter the conclusion.
**Abdominal Obesity**	54 (48–59)	20.073	⊕⊕⊝⊝ ^a,b,c,g,h^ low	The available evidence is sufficient to determine the prevalence, but confidence in the estimate is limited. As more information becomes available, the observed prevalence could change, and this change may be large enough to alter the conclusion.
**High Triglycerides**	- ^x^	4.867	⊕⊝⊝⊝ ^a,b,d,f,h^ very low	The available evidence is insufficient to determine a reliable prevalence, and confidence in the estimate is limited. More information may allow for a more accurate estimation.
**Low HDL-c**	- ^x^	4.605	⊕⊝⊝⊝ ^a,b,d,f,h^ very low	The available evidence is insufficient to determine a reliable prevalence, and confidence in the estimate is limited. More information may allow for a more accurate estimation.
**MetS**	- ^x^	2.604	⊕⊝⊝⊝ ^a,d,e,f,h^ very low	The available evidence is insufficient to determine a reliable prevalence, and confidence in the estimate is limited. More information may allow for a more accurate estimation.
**CI**: confidence interval				
GRADE quality of evidence ratings **High quality**: We are very confident that the effect in the study reflects the actual effect. **Moderate quality**: We are quite confident that the effect in the study is close to the true effect, but it is also possible that it is substantially different. **Low quality**: The true effect may differ significantly from the estimate. **Very low quality**: The true effect is likely to be substantially different from the estimated effect.				

^x^ Because we are very uncertain regarding the effect estimate, we do not present it in Table 7. ^a^ Downgraded by one level (−1) for serious concerns with risk of bias. ^b^ Downgraded by one level (−1) for serious concerns with inconsistency. ^c^ Downgraded by one level (−1) for serious concerns with imprecision. ^d^ Downgraded by two levels (−2) for very serious concerns with imprecision. ^e^ Downgraded by two levels (−2) for very serious concerns with inconsistency. ^f^ Downgraded by one level (−1) for serious concerns with publication bias. ^g^ The observed surrogate endpoints are strongly associated with the outcome, so we did not reduce the evidence. ^h^ Most included studies were cross-sectional, which may lead to further reduction in the quality of the evidence.

## Data Availability

The protocol of the systematic review is available at: https://osf.io/jfm7g (accessed on 2 November 2022).

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
