# Peer review of "Systematic Review and Meta-Analysis of Metabolic Syndrome and Its Components in Latino Immigrants to the USA"

_ijerph, 2023, doi:10.3390/ijerph20021307_

Round 1
Reviewer 1 Report
The article provide an explicit statement of the objective and question review addresses. It follows the guidelines for systematic review and meta-analysis. It is an important problem of public health and there is interest to know the prevalence in immigrant latinos. Good job.
Author Response
We are grateful for your feedback.
Reviewer 2 Report
Abstract – please provide the link to the protocol https://osf.io/jfm7g
Line 127 - Control e Outcomes
Line 240 - thy were pooled
Line 307,308,309 – spacebar - workers[31,33,61,70, workers[44], others[66
There is lack of sensitivity analysis, especually in the case of Fig 2, fig 4, fig 6, fig 9, fig 11 results
Fig 3 – to small font size, barely visible data
Line 430-32 - In the further analyses, a meta-analysis of the prevalence of type 2 diabetes mellitus 430 or high blood glucose was conducted of 7 studies for immigrant latinos and 5 for US- 431
born latinos (Figure) – provide the number of figure
Overall very interesting study, which in my opinion, needs only an update in the field of sensitive analysis for presented outcomes, which could show what factors increased those values. Authors should present how to decrease this value (e.g. eliminating studies based on the number of participants, number of dropouts (if applicable), or many others). Besides that, it was a nice read.
Author Response
We are grateful for your feedback. Following are the answers to your suggestions.
Abstract – please provide the link to the protocol https://osf.io/jfm7g
Answer: The link was included in line 33.
Line 127 - Control e Outcomes
Answer: The correction was made in line 126.
Line 240 - thy were pooled
Answer: The correction was made in line 238.
Line 307,308,309 – spacebar - workers[31,33,61,70, workers[44], others[66
Answer: The corrections were made in lines 309, 310 and 311.
There is lack of sensitivity analysis, especially in the case of Fig 2, fig 4, fig 6, fig 9, fig 11 results
Answer: A paragraph about sensitivity analysis was included in the results (lines 637-652). We also included this information in the method section, in lines 243-248.
Fig 3 – to small font size, barely visible data
Answer: The font size was corrected in the figures.
Line 430-32 - In the further analyses, a meta-analysis of the prevalence of type 2 diabetes mellitus 430 or high blood glucose was conducted of 7 studies for immigrant Latinos and 5 for US-431born Latinos (Figure) – provide the number of figure
Answer: The number was included in line 468.
Reviewer 3 Report
Dear authors
It covers a very wide range of topics. Moreover, while this study has an interesting topic, it needs some revision.
From the beginning of the introduction to line 81, it is general and not very closely related to the background of the study. I hope it will be written more faithfully with the contents related to Latino Immigrants to the US.
Line 140: I recommend replacing it with: "Definition of disease and disease code"
Lines 305-306: 17 articles are written, and the references are sufficiently listed in the table, so there is no need to list all references. There are several places like this throughout the document (eg, line 314). (Look for other MDPI review papers, so please be more polished).
In particular, since the results are explained in figures and tables, it is not necessary to express all references and all numbers.
Please be clear in the results tables, figures and subheadings.
'Latinos in Latin?' and 'Latinos in US?' and 'American born Latino?'
Edit the figures to the same size for easy understanding by the reader.
As there are many references in the author's study, the high risk of cardiovascular disease in Latino immigrant families in the United States has already been studied.
Please make the author's purpose for conducting this research more clear.
In addition, the cardiovascular risk of Latinos has various causes such as diet, culture, and socioeconomic status. Interpretation and approach to this part should be sufficiently discussed.
Author Response
We are grateful for your feedback. Following are the answers to your suggestions.
Line 140: I recommend replacing it with: "Definition of disease and disease code"
Answer: The correction was made in line 139.
Lines 305-306: 17 articles are written, and the references are sufficiently listed in the table, so there is no need to list all references. There are several places like this throughout the document (eg, line 314). (Look for other MDPI review papers, so please be more polished).
In particular, since the results are explained in figures and tables, it is not necessary to express all references and all numbers.
Answer: Since we have many references included in the review, we thought it best to keep the numbers described in the text to make it easier for readers to locate the articles.
Please be clear in the results tables, figures and subheadings.
'Latinos in Latin?' and 'Latinos in US?' and 'American born Latino?'
Answer: We made changes to make them clearer. The changes are highlighted in the tables, figures and subheadings of the results.
Edit the figures to the same size for easy understanding by the reader.
Answer: The font size was corrected in the figures.
As there are many references in the author's study, the high risk of cardiovascular disease in Latino immigrant families in the United States has already been studied.
Please make the author's purpose for conducting this research more clear.
Answer: The purpose of the study is described in lines 108-122.
In addition, the cardiovascular risk of Latinos has various causes such as diet, culture, and socioeconomic status. Interpretation and approach to this part should be sufficiently discussed.
Answer: It was included in lines 711-720.
Round 2
Reviewer 2 Report
I am satisfied with the corrections made.
Author Response
We are grateful for your feedback.
Reviewer 3 Report
Thanks to the authors.
Many improvements have been made. However, I still think that the introduction needs revision.
Up to line 73 is overly general and needs to be shortened.
In addition, it is necessary to mention more about the characteristics of Latinos living in the United States. (e.g., environment, culture, economy, education, home culture, etc.)
Author Response
We are grateful for your feedback. Following are the answers to your suggestions.
We reduced the initial part of the introduction as suggested by the reviewer and included in lines 83-103 some of the issues that are related to the purpose of the research, our results, discussion, and conclusions. We relied on recent Census data to summarize those issues. We understand that we can not address all issues suggested by the reviewer in our systematic review, because it would be necessary for us to write a much larger text to address them in detail.